# Evaluating inositol phospholipid interactions with inward rectifier potassium channels and characterising their role in disease

Tanadet Pipatpolkai [1,2,3], Robin A. Corey[2], Peter Proks[1,3], Frances M. Ashcroft[1,3✉] & Phillip J. Stansfeld [2,3,4✉]

Membrane proteins are frequently modulated by specific protein-lipid interactions. The activation of human inward rectifying potassium (hKir) channels by phosphoinositides (PI) has been well characterised. Here, we apply a coarse-grained molecular dynamics free-energy perturbation (CG-FEP) protocol to capture the energetics of binding of PI lipids to hKir channels. By using either a single- or multi-step approach, we establish a consistent value for the binding of $PIP_2$ to hKir channels, relative to the binding of the bulk phosphatidylcholine phospholipid. Furthermore, by perturbing amino acid side chains on hKir6.2, we show that the neonatal diabetes mutation E179K increases $PIP_2$ affinity, while the congenital hyperinsulinism mutation K67N results in a reduced affinity. We show good agreement with electrophysiological data where E179K exhibits a reduction in neomycin sensitivity, implying that $PIP_2$ binds more tightly E179K channels. This illustrates the application of CG-FEP to compare affinities between lipid species, and for annotating amino acid residues.

[1] Department of Physiology Anatomy and Genetics, Parks Road, Oxford OX1 3PT, UK. [2] Department of Biochemistry, South Parks Road, Oxford OX1 3QU, UK. [3] OXION Initiative in Ion Channels and Disease, University of Oxford, Oxford OX1 3PT, UK. [4] Department of Chemistry, School of Life Sciences, University of Warwick, Coventry CV4 7AL, UK. ✉email: frances.ashcroft@dpag.ox.ac.uk; phillip.stansfeld@warwick.ac.uk

on channels are integral membrane proteins that mediate ionic flux across the plasma membrane. This process can be regulated by the binding of factors such as soluble ligands or phospholipids to the channel. In particular, lipid binding has been shown to affect many types of ion channel, regulating both their oligomeric state and their activation[1]. Impairment of these processes can lead to a range of human and animal diseases. One well-studied class of ion channels are the mammalian inward rectifying potassium (hKir) channels. In the case of Kir6.2, the pore component of the ATP-sensitive potassium ($K_{ATP}$) channel complex, mutations may result in either a loss or gain of channel function, resulting in congenital hyperinsulinism (CHI) and neonatal diabetes (NDM), respectively[2].

Kir channels are activated by phosphoinositides, in particular phosphatidylinositol-4,5-bisphosphate ($PIP_2$)[3]. Different Kir channels exhibit variable affinities and levels of channel activation to different phosphoinositides[4–6]. The $PIP_2$-binding site on Kir channels has been well defined in several crystal structures, such as chicken Kir2.2 [PDB entry: 3SPI][7] and mouse Kir3.2 [PDB entry: 3SYA][8]. Meanwhile, recent advances in cryo-electron microscopy (cryo-EM) have enabled several high-resolution structures of the pancreatic $K_{ATP}$ channel complex (Supplementary Table 1), which comprises a central tetrameric pore formed of Kir6.2 subunits, surrounded by four regulatory sulfonylurea receptor 1 (SUR1) subunits[9]. This octameric complex couples pancreatic β-cell energy status to insulin secretion[10]. Mutations in the Kir6.2 subunit that are located near the $PIP_2$-binding site are associated with NDM (e.g., E179K/A) and CHI (e.g., K67N)[11–13]. A previous study has shown that the K67N mutation does not alter channel surface expression but has reduced channel activation when cell metabolism was inhibited[13]. The mechanism of how E179K/A and K67N mutations affect channel activity is currently unclear.

Kir channels form an attractive target for applying a computational approach to compare binding affinity between phosphoinositides and also assess the impact of mutations on binding. Coarse-grained (CG) molecular dynamics (MD) simulations have previously been used to identify lipid-binding sites on ion channels[14–16], as well as predicting the affinity of interactions[17].

For many years, the application of atomistic free energy perturbation (FEP) methods have been successfully applied to determine small molecule, lipid, and drug-binding affinities[18], as well as to study the impact of amino acid side chain mutations[19,20]. Our recent study showed how the method could be extended to a CG protocol (CG-FEP) to assess relative protein–lipid binding free energies. This approach was in strong agreement with other free energy calculation methods such as potential of mean force (PMF) calculation and well-tempered metadynamics[17].

In this study, we use CG-FEP[17] to compare the relative binding free energies between different phospholipids and the human Kir6.2 channel, capturing the full thermodynamic cycle for the transition of $PIP_2$ to PC (1-palmitoyl-2-oleoyl-sn-glycero-3-phosphocholine), either directly or via intermediates phosphatidylinositol-4-phosphate (PI4P) and phosphatidylinositol (PI), and thereby reporting on the affinity of each interaction. We extend the methodology to investigate the functional effect of lipid-associated NDM mutations (E179K/A) and a CHI mutation (K67N) in hKir6.2[21,22]. Based on the predicted binding site for $PIP_2$, we calculate that these residues interact with $PIP_2$ in the membrane. This therefore provides a biochemical and structural explanation for the different clinical phenotypes.

We couple these analyses with electrophysiology, to assess both the affinity for, and channel activation by, $PIP_2$. We also extend the computational methodology to assess the binding free energy differences between a range of hKir channels (hKir1.1, hKir2.2,

and hKir3.2) and other inositide lipid species such as PI4P and PI. Together, our application of the CG-FEP describes the affinity of membrane proteins with a range of different lipids, as well as examining how biologically important mutations affect these interactions.

## Results

**$PIP_2$ binding conformation to the hKir6.2 channel**. The initial position of the $PIP_2$ molecule was obtained from the chicken Kir2.2 channel:diC8-$PIP_2$ complex[7]. After structural alignment of hKir6.2 with chicken Kir2.2, one of the bound diC8-$PIP_2$ molecules was extracted, converted to CG, and the resultant hKir6.2-$PIP_2$ complex was built into a PC membrane and simulated for 1 μs ($n = 5$) using CG lipid self-assembly[14,15,23]. We defined residues that were within a 6 Å radius of the whole $PIP_2$ molecule for >75% of simulation time as proximal residues (Fig. 1a). We found that $PIP_2$ binds in the vicinity of both the N and C termini of the hKir6.2 channel, including ⁶⁷KWP⁶⁹ on the N terminus and the residues between 170 and 179 on the C terminus. These regions contain a number of basic residues, which allows them to interact with the negatively charged phosphate groups on the inositol ring of the $PIP_2$ headgroup. E179 is the only negatively charged amino acid to be within this cut-off from the lipid.

We assessed the stability of $PIP_2$ in its binding site using root mean square deviation (RMSD) analysis over the 1 μs simulations (Fig. 1b). The data show that the position of the $PIP_2$ diverges very little in 1 μs (RMSD = ca. 0.8 Å). This was corroborated by analysing the distance between the $PIP_2$ molecule and two amino acids near the $PIP_2$-binding site, K67 and E179. We found that the minimum distance between K67 and E179, and the $PIP_2$ headgroup are ~5 Å (Fig. 1c, d) and ~7 Å between lipid and protein backbone (Supplementary Fig. 1). Therefore, we hypothesise that mutations to these residues may affect the binding free energy of $PIP_2$ to the channel.

**Stepwise perturbation of the PIP molecule bound to the hKir6.2 channel**. We next assessed the contribution that each of the different PIP headgroup moieties (i.e., each phosphate group and the inositol ring) make to the free energy of binding to the closed hKir6.2 tetramer using CG-FEP. For this, we iteratively perturbed single beads to transform from one phospholipid (such as $PIP_2$) into another (such as PC). This enabled us to calculate the binding free energy difference ($\Delta\Delta G$) of the two different phospholipids to hKir6.2, embedded in a PC bilayer (Fig. 2a). For simplicity, all of our lipids have both palmitoyl and oleoyl alkyl chains. The energies were computed using Multistate Bennett Acceptance Ratio (MBAR)[24], with convergence seen within ca. 200 ns per window (Supplementary Figs. 2a, b, 3a, b, 4a, b, and 5a, b). To prevent the molecule from leaving its binding site, we applied a flat-bottom distance restraint between the protein and lipid using Plumed[25,26] (Supplementary Figs. 2a–c and 3a–c). This was mostly applicable for the calculations in which the lipid was transformed to PC. In the cases where the lipids remain bound at the binding site, applying a flat-bottom restraint makes no difference to the binding free energy and its convergence. (Supplementary Figs. 4c and 5c). This procedure also reduced the errors between simulation replicas. The infrequency with which the lipid experiences the restraint suggests that it has negligible effect on the binding energies (Supplementary Figs. 2d and 3d).

Transformation from $PIP_2$ to PI4P showed a very small relative free energy change (Fig. 2b and Supplementary Table 2). This suggested that a phosphate group at the 5′-position does not make a considerable contribution to $PIP_2$ binding (Fig. 2b and Supplementary Table 2). However, we observed large free energy changes when the phosphate group at the 4′-position and the

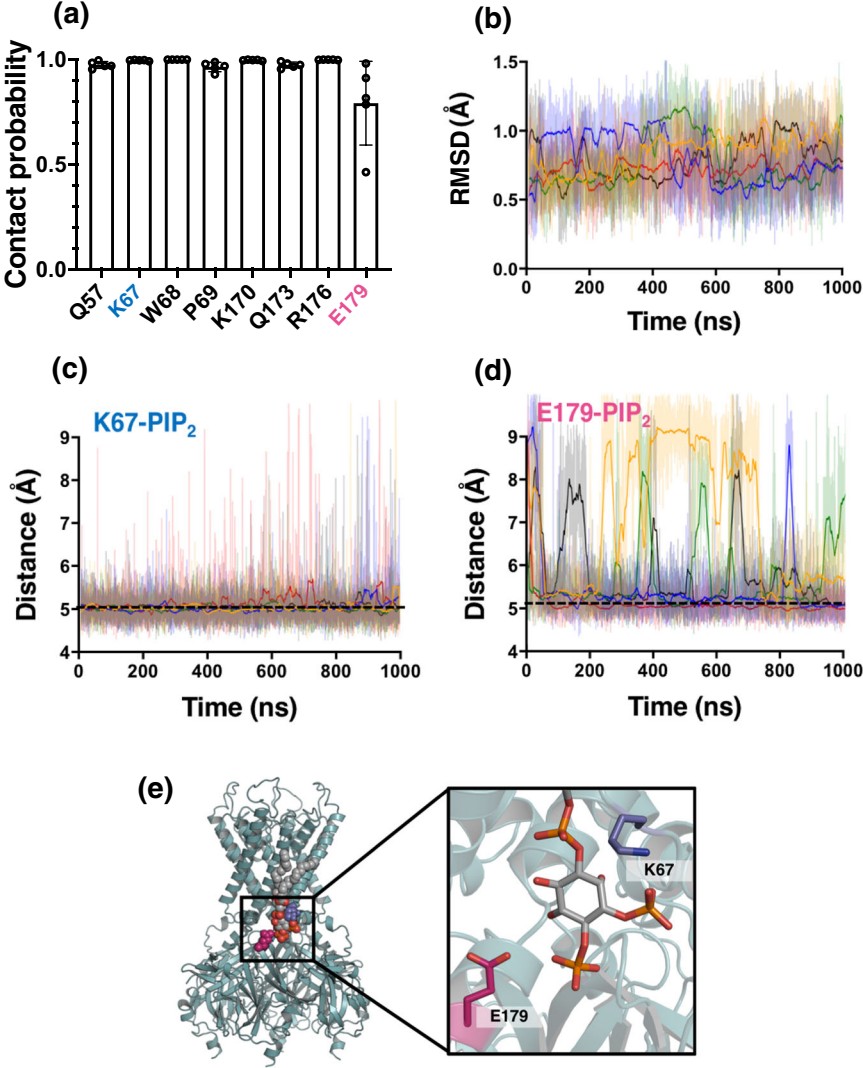

**Fig. 1 A PIP$_2$-binding site on hKir6.2. a** PIP$_2$ contact analysis showing the fraction of time that residues are in 6 Å proximity to the PIP$_2$ molecule (contact probability). Only residues with a >75% contact probability are shown. Data from five repeats of 1 μs simulations. **b** Root mean square deviation (RMSD) analysis of the PIP$_2$ molecule headgroup when bound to the hKir6.2 tetramer. The different colours indicate the individual repeats of the simulations (*n* = 5). The darker lines show the running average for each simulation. **c** Calculated minimum distance between K67 and the PIP$_2$ headgroup during a 1 μs simulation. The different colours indicate the individual repeats of the simulation. The darker line shows a running average. A dashed black line denotes the distance cut-off used to denote a contact in **a**. **d** As in **c** but for E179 and the PIP$_2$ headgroup. **e** PIP$_2$-binding site on the hKir6.2 tetramer (green) showing PIP$_2$ (grey with CPK colours), E179 (magenta), and K67 (blue).

inositol ring were perturbed (Fig. 2b and Supplementary Table 2). Summation of each individual moiety from PIP$_2$ to PC gives a binding free energy of 30 ± 1 kJ/mol, which is remarkably similar to the 33 ± 3 kJ/mol we obtain for direct perturbation of PIP$_2$ to PC (Fig. 2c, Supplementary Table 2, and Supplementary Fig. 6). This suggests that the approach we use is valid for both single- and multi-step free energy calculations.

A previous study demonstrated that crosstalk between different anionic lipids can affect the affinities of each lipid for the Kir2.2 channel[27]. Here we show that the presence of 10% anionic lipid - phosphatidylserine (PS) in the lower leaflet of the bilayer does not affect the overall ΔΔG for PIP$_2$ binding to Kir6.2 (Supplementary Fig. 7).

Previous electrophysiological and crystallographic studies have commonly used the soluble eight-carbon atom phosphatidylino-sitol, diC8-PIP$_2$, to study channel activation[7,8,12]. Therefore, we investigated the effect of the length of the acyl chain on PIP$_2$-binding affinity. We found that truncation of the acyl chain from

either four or five particles (i.e., palmitoyl and oleoyl) to two particles (equivalent to eight carbon atoms) had no effect on PIP$_2$ affinity (Fig. 2d). This suggests that PIP$_2$-diC8 is indeed an effective substitute for investigating the impact of PIP$_2$ binding in electrophysiological and structural studies.

**Relative binding free energy calculations for PIP2 interactions with both wild-type and mutant hKir6.2.** Based on the closed state model of Kir6.2 [PDB: 6BAA], we generated three structural models of hKir6.2 with disease-associated mutations: K67N, which causes CHI, and E179A and E179K, which cause NDM[21,22]. As these residues are in close proximity to the PIP$_2$-binding site (Fig. 1e), we hypothesised that mutations to these residues would modulate PIP$_2$ affinity and thereby affect basal channel activity (i.e., the channel open probability, Po). An increase in PIP$_2$-binding affinity should correlate with an increase in channel activation (Po) and thus also a reduced inhibition by ATP.

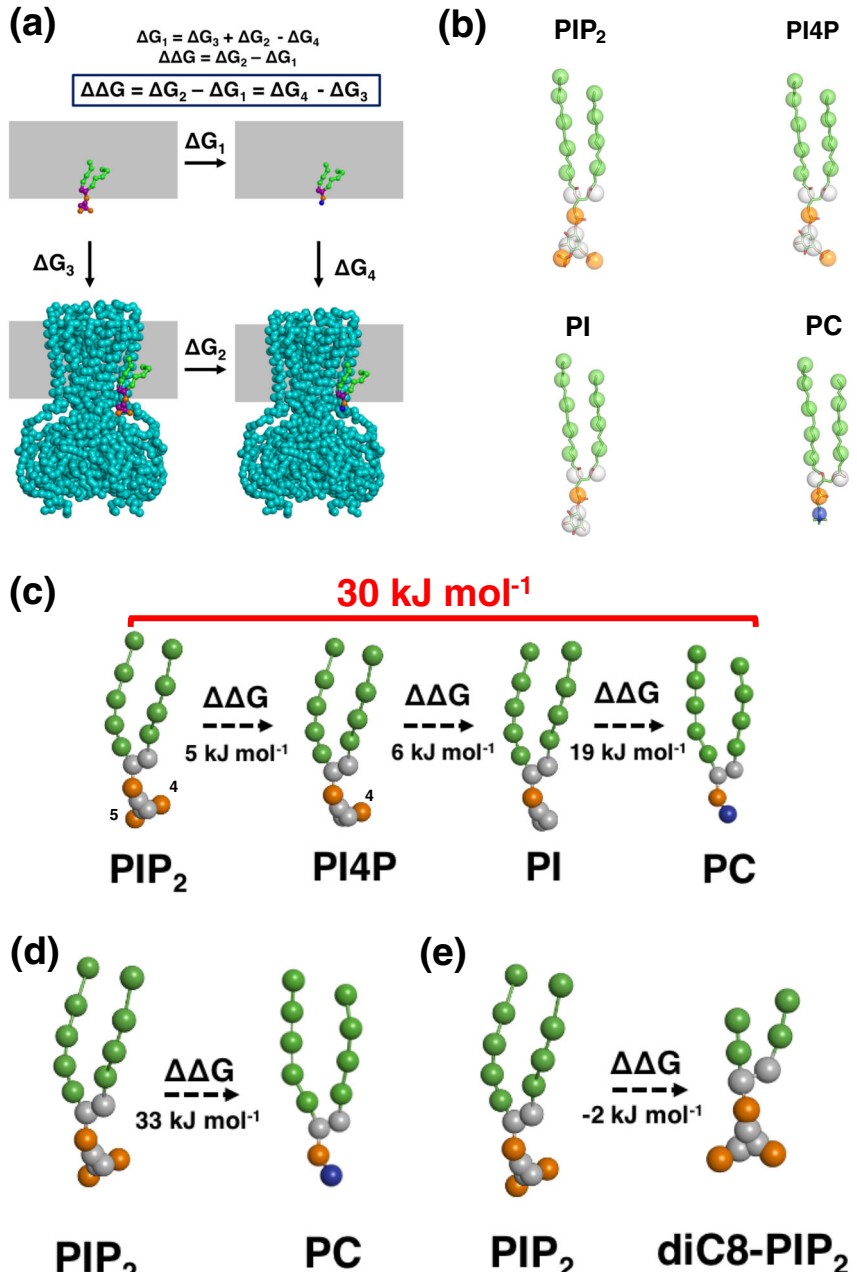

**Fig. 2 The free energy calculation of an individual phosphate group and fatty acid chains on a hKir6.2 tetramer. a** Thermodynamic cycle used for the relative binding free energy calculations. The perturbation of the $PIP_2$ headgroup (purple) was calculated in both the channel-bound state ($\Delta G_2$) and free in the PC membrane (grey rectangle ($\Delta G_1$)). $\Delta G_3$ and $\Delta G_4$ can be calculated using such methods as PMF calculations. **b** Coarse-grain to atomistic mapping of the phosphoniosites ($PIP_2$, PI4P, PI, and PC). **c** Change in binding free energy ($\Delta\Delta G$) when individual phosphate groups are perturbed (i.e., from $PIP_2$ to PI4P, from PI4P to PI and from PI to PC (values in black). The sum of these free energy changes (i.e. from $PIP_2$ to PC) is given in red. Values are rounded to the nearest whole number. **d** Change in binding free energy when $PIP_2$ is perturbed to PC. Values are rounded to the nearest whole number. **e** Change in binding free energy when $PIP_2$ is perturbed to $PIP_2$-diC8. Values are rounded to the nearest whole number.

We next performed calculations in which the mutated hKir6.2 residue was perturbed in the presence and absence of $PIP_2$ (Fig. 3a). This allows us to calculate the relative changes in the $PIP_2$-binding free energy between the wild-type and mutant channels. For the highest energy residue substitution (E179K), we observed convergence of the free energy calculations within 50 ns per window (Supplementary Fig. 8). The data show an increase in $PIP_2$-binding energy, and hence an increased affinity, with the E179K transformation. We also observed an increase in binding free energy with the E179A mutation (Fig. 3b and Supplementary Table 3). Conversely, we see a reduction in $PIP_2$-binding affinity

with the K67N transformation (Fig. 3b and Supplementary Table 3). This quantitatively confirms that both the E179K and E179A mutations increase $PIP_2$ channel affinity, whereas the K67N mutation decreases channel affinity. This agrees with the patient phenotypes: E179K causes NDM, i.e., an increase in channel activity, whereas K67N causes CHI, a reduction in channel activity.

As a control, we investigated an NDM mutation, C166S, which is distant from the $PIP_2$-binding site. We expected that this mutation would have no impact on the $PIP_2$-binding affinity, despite having an influence on the channel opening probability[28].

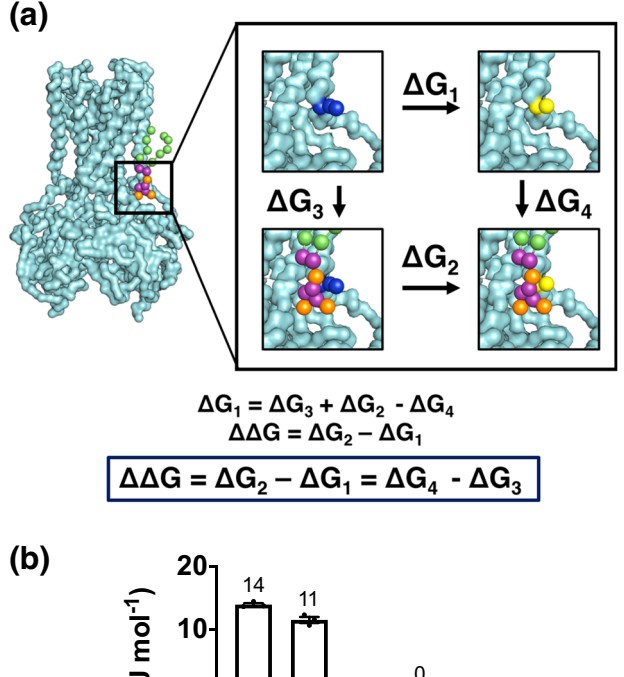

$$\Delta G_1 = \Delta G_3 + \Delta G_2 - \Delta G_4$$
$$\Delta\Delta G = \Delta G_2 - \Delta G_1$$

$$\boxed{\Delta\Delta G = \Delta G_2 - \Delta G_1 = \Delta G_4 - \Delta G_3}$$

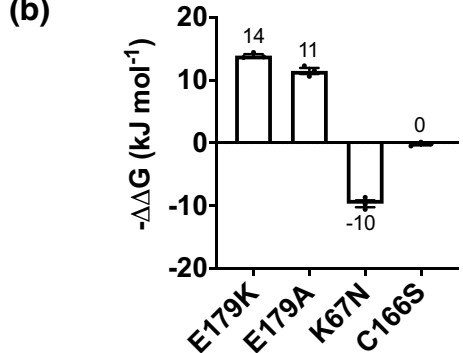

**Fig. 3 Free energy calculations using disease-associated Kir6.2 mutations. a** Schematic diagram showing the free energy calculation. An amino acid residue (blue spheres) is transformed into another residue (yellow spheres) between two states (PIP$_2$ bound and free). **b** The energetic cost of making the residue mutation based on the schematic diagram (**a**).

When perturbing the site using single residue FEP, we see effectively no change in PIP$_2$ affinity of the channel, and therefore, unlike the E179A/K mutations, the C166S mutant does not appear to increase channel opening probability by increasing PIP$_2$ affinity (Fig. 3c and Supplementary Table 3), but by a different mechanism.

**Experimental assessment of PIP2 binding to NDM mutant channels**. We next assessed the neomycin sensitivity of the E179K mutant both in the presence and the absence of SUR1 to ascertain whether the mutation increases both channel PIP$_2$ affinity and channel activation (Fig. 4a, b). To enable expression of Kir6.2 without SUR1, we used a C-terminally truncated construct, Kir6.2ΔC36, which has been previously shown to express and traffic to the plasma membrane without SUR1[29]. Neomycin, a polycationic antibiotic, has previously been used as a tool to study Kir ion channel activation by PIP$_2$[12]. Although it does not bind to Kir channels directly, it acts by reversibly binding to PIP$_2$, screening the charges, and preventing binding[30]. A decrease in neomycin sensitivity (i.e., an increase in neomycin IC$_{50}$) would indicate an increase in PIP$_2$ dependent channel activation (i.e., PIP$_2$ has a greater affinity for the Kir channel upon mutation). Another indication of an increase in channel activation by PIP$_2$ is a slower rate of channel rundown and an increase in channel

open probability[31]. We used these criteria to assess the effect of the SUR1 subunit on channel sensitivity to PIP$_2$.

We observed a 20-fold and a 50-fold increase in the IC$_{50}$ value for neomycin block with the E179K mutation, both in the presence and absence of SUR1, respectively (Kir6.2-SUR1: IC$_{50}$ = 81 μM, $h = 0.96$; Kir6.2-E179K-SUR1: IC$_{50}$ = 1.7 mM, $h = 1.13$; Kir6.2ΔC, IC$_{50}$ = 55 μM, $h = 0.61$; and Kir6.2ΔC-E179K, IC$_{50}$ = 3.3 mM, $h = 1.6$). This suggests an increase in the PIP$_2$ sensitivity of the E179K variant channel, both in the presence and in the absence of the SUR1 subunit, in agreement with our free energy calculations. Interestingly, in the presence of SUR1, channels with the Kir6.2-E179K mutation failed to fully close even in the presence of a very high neomycin concentration (0.1 M) (Fig. 4b). This suggests that the E179K mutation, in the presence of SUR1, may interfere with the channel-gating mechanism independently of PIP$_2$ action.

**Assessment of the PIP$_2$ activity dependency on SUR1 subunit**. Next, we experimentally assessed the importance of the SUR1 subunit for PIP$_2$-binding affinity and activation. Previous studies suggested that the presence of SUR1 enhances the Po of Kir6.2ΔC[32,33]. However, the contribution of PIP$_2$ to this modulation and the relationship between SUR1 and PIP$_2$ sensitivity remains unclear. To address this issue, we calculated the relative binding free energy of hKir6.2 and PIP$_2$ in both the presence and absence of the SUR1 subunit. Here we show that addition of SUR1 only marginally increases the PIP$_2$-binding free energy (Fig. 4c, Supplementary Table 4). Therefore, this result suggests that SUR1 has only a minor contribution to PIP$_2$ affinity even though the SUR1 is only ~8 Å away from PIP$_2$ headgroup.

To confirm that SUR1 has no effect on channel activation, we cloned and expressed Kir6.2 with SUR1 and Kir6.2ΔC in *Xenopus* oocytes, and assessed the neomycin sensitivity of the channel. We found there was no significant difference in channel neomycin sensitivity in the presence and absence of SUR1 (Kir6.2/SUR1: IC$_{50}$ = 81 μM, $h = 0.96$; Kir6.2ΔC, IC$_{50}$ = 54 μM, $h = 0.61$) (Fig. 4d). This is in qualitative agreement with the lack of a change in the free energy of PIP$_2$ binding calculated from our FEP calculations.

**PIP$_2$-binding affinity to other hKir channels**. We next assessed the binding of PIP$_2$ across other hKir channels by calculating the binding free energy of PIP$_2$ perturbation to PC for the human Kir1.1, Kir2.2, and Kir3.2 (hKir1.1, hKir2.2, and hKir3.2) channels using the thermodynamic cycle described in Fig. 2a. The electrophysiological behaviour of these channels on PIP$_2$ binding is well characterised[4]. We therefore perturbed PIP$_2$ to PC, the dominant phospholipid species in the eukaryotic plasma membrane. It is noteworthy that a PMF calculation shows the binding energy of PC to hKir6.2 is $0 \pm 2$ kJ/mol (Supplementary Fig. 9). Due to the absence of human Kir1.1 and human Kir2.2 structures in the PIP$_2$-bound conformation, we generated hKir molecular models of both, as described in the "Methods." The PIP$_2$-binding sites and interacting residues on these proteins are all highly conserved (Fig. 5a–d and Supplementary Fig. 10).

Our data reveal that the binding free energy of PIP$_2$ to hKir1.1, hKir2.2, and hKir3.2 channels is higher than that to hKir6.2 channels (Fig. 6a and Supplementary Table 5). In our study, the PIP$_2$-binding free energy value described here for hKir2.2 is similar to that previously recorded for the chicken Kir2.2 channel, which was reported as ca. $-46$ kJ/mol, both using FEP and PMF[17,34]. The higher affinity of hKir3.2 was rather unexpected, as hKir3.2 has an equivalent glutamate in the same position at E179 on hKir6.2 (denoted as E201) (Fig. 5d). However, the minimum distance between the PIP$_2$ headgroup and hKir3.2-E201 (ca. =

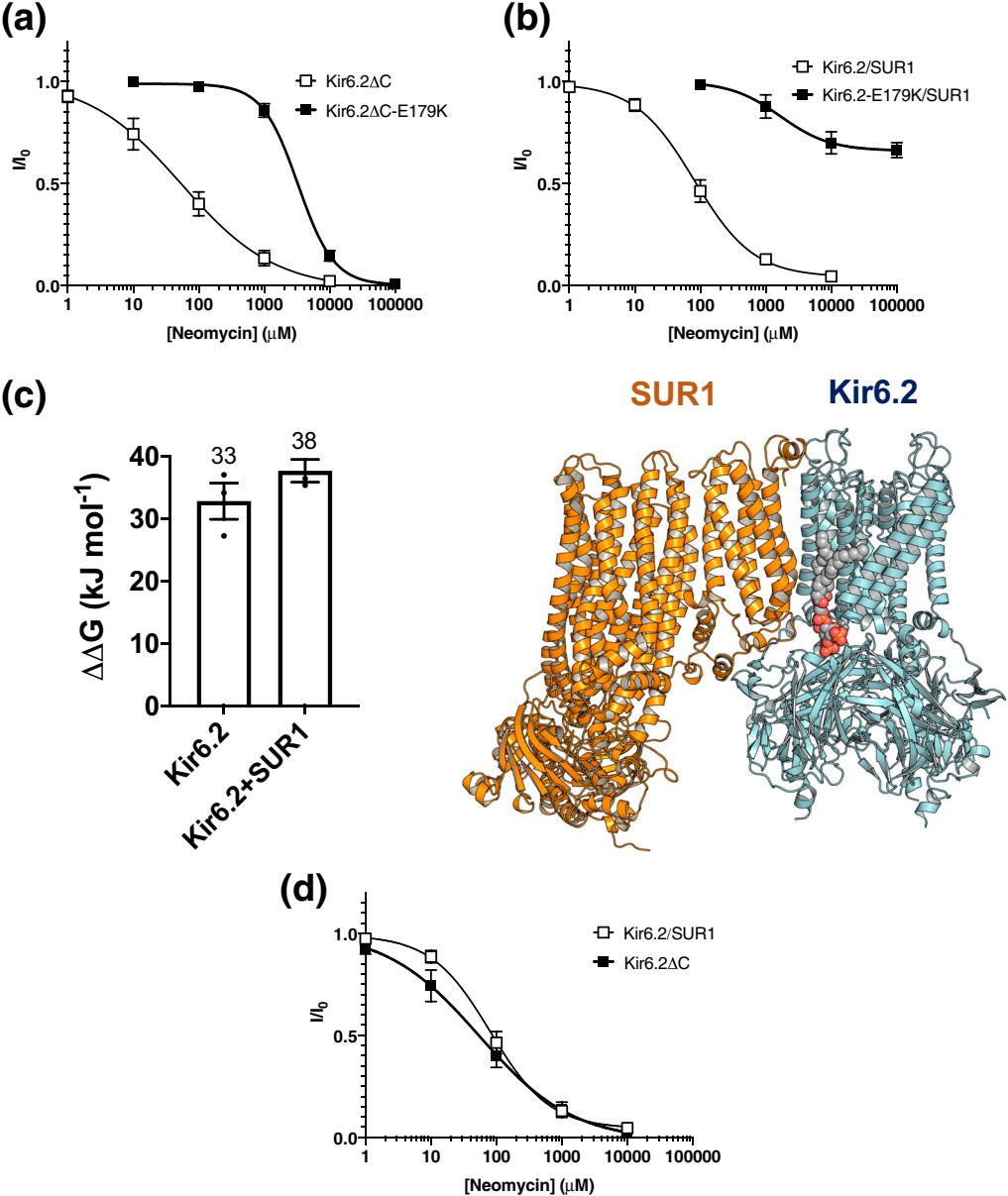

**Fig. 4 An effect of SUR1 subunit and NDM mutation on PIP$_2$ affinity and activation. a** Mean relationship between the neomycin concentration and the K$_{ATP}$ current (*I*), expressed relative to the current in the absence of neomycin (*I$_0$*) for Kir6.2ΔC (open squares, *n* = 5) or hKir6.2ΔC-E179K channels (filled squares, *n* = 5). **b** Mean relationship between the neomycin concentration and the K$_{ATP}$ current (*I*) expressed relative to the current in the absence of neomycin (*I$_0$*), for Kir6.2/SUR1 (open squares, *n* = 5) or hKir6.2-E179K/SUR1 channels (filled squares, *n* = 5). **c** Left: binding free energy between PIP$_2$ and Kir6.2 ± SUR1. Values are rounded to the nearest whole number. Error bars represent the SEM (*n* = 3). Right: the PIP$_2$ (grey) binding sites between hKir6.2 and SUR1. **d** Mean relationship between the neomycin concentration and the K$_{ATP}$ current (*I*), expressed relative to the current in the absence of neomycin (*I$_0$*), for Kir6.2 co-expressed with SUR1 (open squares, *n* = 5) or hKir6.2ΔC expressed without SUR1 (filled squares, *n* = 5).

8.5 Å) is greater than that of hKir6.2-E179 (ca. = 5 Å) in our molecular models.

To account for the free energy difference between PIP$_2$ and PC to all hKir channels, we examined the contribution of the individual phosphate and inositol groups of PIP$_2$ in the binding to each of the channels, as described above (Fig. 2a). First, we show that the binding energy contributed by the 5′-phosphate is higher in hKir1.1 than the other three channels (PIP$_2$ > PI4P; Fig. 6b and Supplementary Table 6). In addition, we observed that 4′-phosphate contribution is stronger in all other channels relative to hKir6.2 (PI4P > PI; Fig. 6b and Supplementary Table 6). Overall, this accounts for most of the energy differences as we perturbed PIP$_2$ to PC in hKir1.1, hKir2.2, and hKir3.2. For both

single- and multi-step approaches, hKir6.2 exhibited the lowest binding free energy for PIP$_2$.

## Discussion

Here we build on our recent application of the CG-FEP approach for comparing the binding free energy between two lipid species to a given site on a membrane protein[17]. We show that this approach enables us to complete a thermodynamic cycle where the sum of the individual perturbation steps (i.e., PIP$_2$ > PI4P > PI > PC) is equivalent to the single-step transformation from PIP$_2$ to PC (Supplementary Fig. 6). The results of our application of the free energy calculations are within the range of values that are

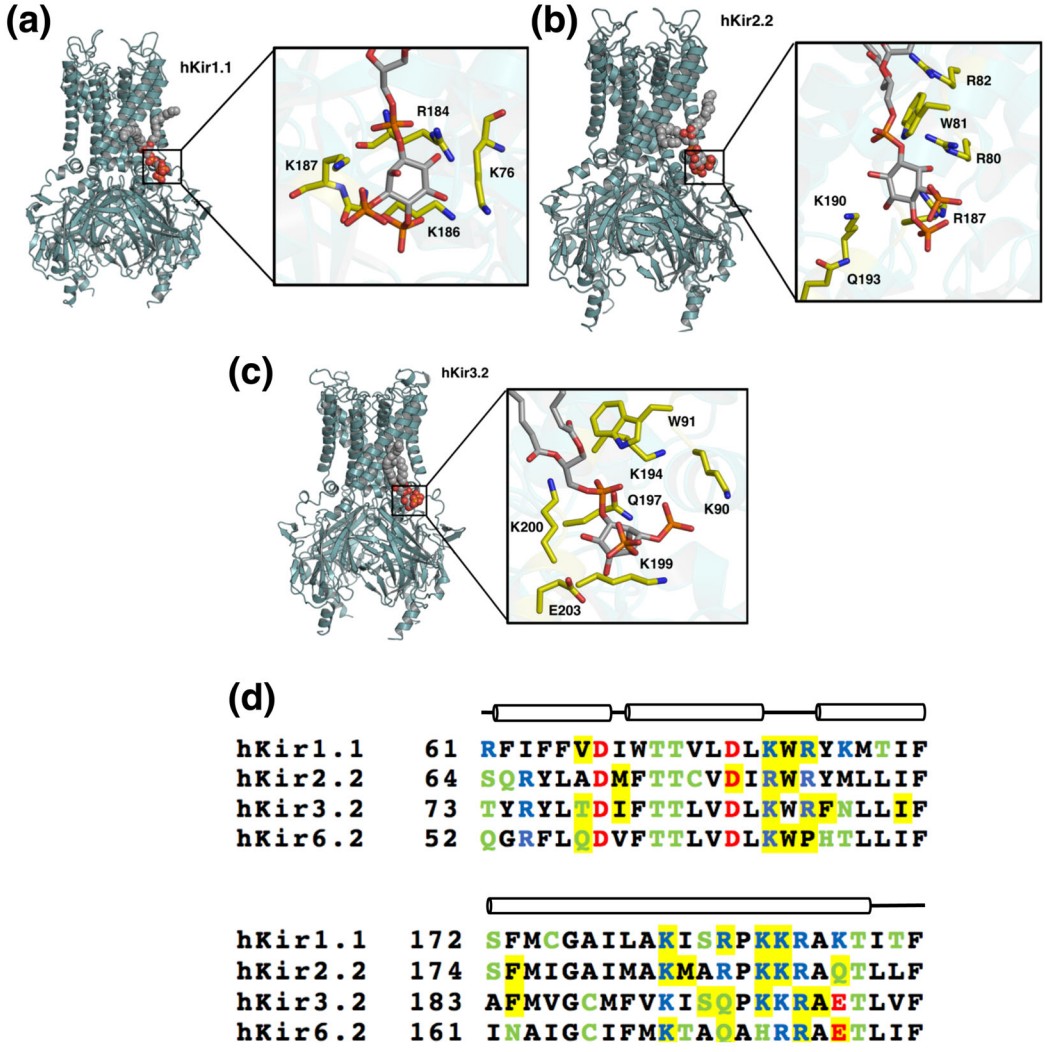

**Fig. 5 Free energy calculations for different hKir channels. a** Models of a hKir1.1, **b** hKir2.2, and **c** hKir3.2 channels in the PIP₂-bound conformations after 1 μs of CG simulation and converted back to an atomistic description. Insets: carbons of key PIP₂-binding residues are highlighted in yellow, with PIP₂ otherwise shown in CPK colours. **d** Sequence alignment between hKir1.1, hKir2.2, hKir3.2, and hKir6.2 channels on the region where the contacts are conserved between more than two channels. Highlighted in yellow are residues that contact PIP₂ for >70% of the 1 μs simulations ($n = 5$). Long cylinder represents an α-helix in the secondary structure and the line represents either disordered region or a kink within the α-helix. Acidic residues (Asp, Glu) are shown in red, basic residues (Lys, Arg) are shown in blue, and polar residues (His, Ser, Thr, Cys, Gln, Asn) are shown in green. Other residues are shown in black.

commonly observed for PIP₂ interactions with membrane proteins[17,34–36]. One of our concerns is the ability of the CG forcefield to distinguish between similar inositol lipids in the free energy calculations. This application of CG-FEP has demonstrated that the method can show differences between PIP₂, PI4P, PI and PC at the accuracy of at least 5 kJ/mol (~1.5 $k_B T$). This complements the previous free energy calculations, which show that the CG forcefield is able to distinguish between PIP₂ and PIP₃[36]. Overall, this illustrates the robustness of the application and demonstrates its power as a relatively cheap and effective in silico approach for comparing lipid-binding free energies to a membrane protein of interest. Nevertheless, although there is good agreement between the single- and multi-step approaches we cannot exclude the possibility that the individual particle contributions may be either over- or under-estimated due to the CG approach.

In addition, we demonstrate that amino acid mutations can also be investigated using CG-FEP, thus allowing us to probe the effect of a given amino acid's substitution on lipid binding. This is

an extension to the traditional atomistic FEP mutation approach[20], which enables convergence to be achieved more quickly and easily[20]. It also allows binding to be much more easily measured than in vitro lipid-binding studies. Thus, the application of CG-FEP is potentially a valuable tool for the relatively high-throughput analysis of multiple disease-causing mutations that are related to lipid binding.

To test the capabilities of these methodologies, we applied them to the K_ATP channel (Kir6.2/SUR1), a biologically important potassium channel that is implicated in insulin secretion. We used a combination of the above methods to demonstrate that clinically identified mutations in the Kir6.2 subunit (causing NDM and CHI) affect the affinity of the channel for different PIP lipids. In addition, we demonstrated that the NDM mutations—E179K and E179A—which lie near the PIP₂-binding site result in channel gain of function by enhancing PIP₂ binding (and Po). We present both computational and electrophysiological data, which are in good agreement, thereby demonstrating that this application of an existing method provides a potentially powerful

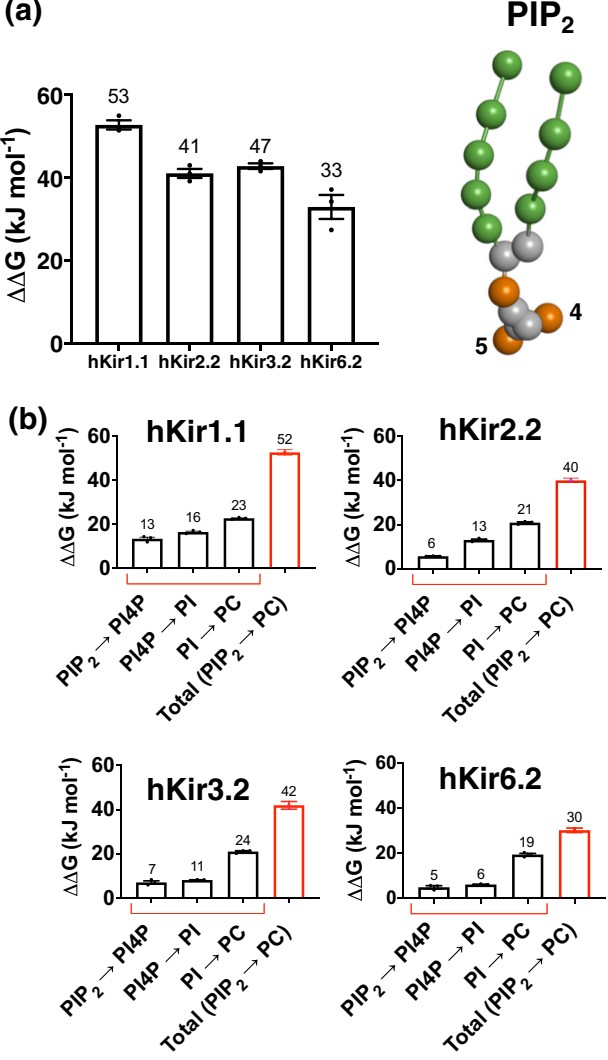

**Fig. 6 Free energy calculations for different hKir channels. a** Binding free energy changes between different hKir channels as PIP$_2$ is perturbed to PC ($n = 3$). **b** Binding free energy changes between PIP$_2$ and hKir1.1, hKir2.2, hKir3.2, or hKir6.2, as each PIP$_2$ phosphate group is sequentially perturbed: from PIP$_2$ to PI4P, then to PI and finally to PC (black). The sum of the free energy change from PIP$_2$ to PC is shown in red. Values are rounded to the nearest whole number. Error bars represent the SEM ($n = 3$).

method for the scanning and annotating of how disease-associated mutations modulate lipid binding to channels.

## Methods

**Molecular modelling.** Modeller9v16[37] was used to add the missing loops and amino acid residues to the cryo-EM structure of the Kir6.2 channel [PDB entry: 6BAA][38] and to generate a human Kir6.2 model based on residues 32–352. In both the cryo-EM structure and our model there are 32 amino acids missing at the N terminus and 39 at the C terminus. Modeller9v16 was also used to generate a model of the human K$_{ATP}$ channel octamer (hKir6.2 tetramer + four SUR1 [PDB entry: 6BAA]) and the models of the hKir6.2 mutant channels. Each model was compared to its initial template structure to ensure that the modelling had not demonstrably altered the original secondary structure or the rotation of the amino acid side chains (overall RMSD of all protein atoms <1.0 Å). Models of the other hKir channels were generated using Swiss-Model[39], with human Kir1.1 and Kir2.2 based on 3SPH and human Kir3.2 based on 3SYA. The chicken Kir2.2 structure with bound diC8-PIP$_2$ [PDB entry: 3SPI][7] was used to dock PIP$_2$ to hKir6.2.

**CG simulations.** All protein structures were converted to their CG representation and embedded in a PC bilayer using the self-assembly MemProtMD protocol[23,40] and the MARTINI v2 biomolecular forcefield[15]. This approach orients the

structure of the transmembrane protein parallel to the z-axis using MEM-EMBED[41]. The protein is then placed in a periodic box at minimum distance of 30 Å from the edge of the box in both x and y directions, and with a z-dimension of 80 Å. The structure is then converted to a CG representation with *martinize.py* with an application of an elastic network with a force constant of 1000 kJ/mol/nm$^2$ between backbone beads within 0.5–0.9 nm to maintain their secondary and tertiary structure. The PC lipid is then added to the periodic box, allowing them to assemble freely around the protein. The z-dimension of the box is then extended so that the minimum distance between the protein and the face of the box is 30 Å apart, and then flooded with the coarse-grain water particles, Na$^+$ and Cl$^-$ ions to a final concentration of 0.15 M to neutralise the system. The total number of the molecules in the setup is described in the Supplementary Table 7. A temperature of 323 K was maintained with V-rescale temperature coupling[42], whereas 1 atm pressure was controlled using semi-isotropic Parrinello–Rahman pressure coupling[43]. Systems were energy minimised using the steepest descents algorithm and equilibrated for 5 ns with 1000 kJ/mol/nm$^2$ position restraints on backbone beads, prior to 1 μs production. All simulations, RMSD calculations and distance analyses were carried out using GROMACS v2018[44] and all structural alignments and docking were carried out using PyMOL[45].

**FEP calculation of PIP lipids.** The hKir6.2 tetramer with one bound PIP$_2$, obtained after equilibration, was used as the initial co-ordinates for the majority of the FEP calculations. Here we calculate a relative binding free energy ($\Delta\Delta G$) by converting from one lipid type (such as PIP$_2$) to a series of other phospholipids (such as PIP$_2$, PI4P, PI, and PC) along a reaction co-ordinate in a chemical space denoted λ (Fig. 2a). As is standard for FEP calculations, separate transformations were performed with either the lipid bound to the channel or in bulk membrane.

We applied FEP to hKir6.2, hKir3.2, hKir2.2, and hKir1.1, and the following pairs of inositol lipids: (PIP$_2$ and PI4P), (PI4P and PI), (PI and PC), and (PIP$_2$ and PC). This allows us to create a thermodynamic cycle for the different lipids of interest (Supplementary Fig. 6). For these, specific phosphate and inositol sugar particles were transformed into a dummy particle with no interaction properties, in a stepwise process as described in Fig. 2b. Coulombic (charge interactions) and Lennard–Jones (van der Waals interactions) were turned off separately, with a soft-core parameter used for the Lennard–Jones interactions. The coulombic interactions were perturbed linearly (λ = 0, 0.1, 0.2 … 0.9, 1.0) in the first ten simulation windows, with the van der Waals interactions perturbed linearly (λ = 0, 0.1, 0.2 …… 0.9, 1.0) in the last ten simulation windows with the soft-core parameters of α = 0.5 and σ = 0.3. Each simulation window was energy minimised and equilibrated as described above, before three production runs were carried out for 300 ns with randomised initial velocities, using a leap-frog stochastic dynamics integrator. A flat-bottom distance restraint between the PO4 phosphate group and protein backbone beads at 6 Å radii from the PIP molecule was applied using Plumed (1000 kJ/mol/nm$^2$, 8 Å cut-off)[26]. This prevents the bound lipid from drifting away from its binding pocket and increases the accuracy of the calculation (Supplementary Fig. 1b, c). The free energy pathways were constructed using the *alchemical-analysis* software package[46], where the energies are calculated based on the 300 ns of the data for a good convergence[17]. Thus, a total of 642.6 μs simulations were performed for the FEP calculations performed in this study. Analyses was run using the MBAR[24]. All values are reported are reported as mean ± SEM. All simulations were carried out using GROMACS v2018[44].

**FEP between amino acid residues.** As before, the PIP$_2$-bound equilibrated hKir6.2 system was used as the initial co-ordinates for the FEP calculations. When assessing the influence of ND or CHI mutations, we calculated changes in the relative binding free energy ($\Delta\Delta G$) by alchemically transforming the wild-type amino acid residue to its mutant counterpart. This was performed using a change in chemical space denoted as λ, and applying the previously described protocol. The series of transformations carried out are shown in Supplementary Table 8. Additional simulations were run by performing the same perturbation of the protein in a bulk phosphatidylcholine (PC) membrane in the absence of PIP$_2$. The $\Delta\Delta G$ terms were then calculated as described in Fig. 3b.

**PMF calculation.** PMF calculations were set up similarly to that described previously[17]. The protein is built in the PC bilayer, and a single PC lipid is pulled from the binding site using steered-MD, where the collective variables (CV) are the distance between the lipid headgroup and the centre of mass of the protein. The initial position for PC was modified from an initial PIP$_2$ co-ordinate. The simulations were calculated along the CV at 0.2 Å interval for optimal histogram overlap, with a 1000 kJ/mol/nm$^2$ umbrella potential applied to restrain the position of the lipid along the CV. Positional restraints of 100 kJ/mol/nm$^2$ were applied to the protein backbone to prevent rotation of the protein in the bilayer. For each window, the simulations were run for 500 ns, which was sufficient to see convergence. Thus, this adds up to a total of 20 μs for the PMF calculations. The one-dimensional energy profile was generated using weighted histogram analysis method using the *gmx wham* tool (200 rounds of Bayesian Bootstrap)[47].

**Molecular biology.** Human Kir6.2 (Genbank NM000525) and human SUR1 (Genbank NM_000352.5) were cloned into the pBF vector. Site-directed

mutagenesis was performed using QuickChange XL (Stratagene), followed by synthesis of capped mRNA using mMESSAGE (Invitrogen). All constructs were validated by restriction digest and DNA sequencing (MRC I PPU, School of Life Science, University of Dundee, Scotland). *Xenopus laevis* oocytes were prepared as previously reported[48]. The oocytes were co-injected with ~4 ng of SUR1 mRNA and ~0.8 ng wild-type or mutant Kir6.2 mRNA. In some experiments, oocytes were injected with wild-type or mutant Kir6.2 possessing a C-terminal 36 amino acid truncation (Kir6.2ΔC) mRNA, which allows surface membrane expression[32]. Oocytes were incubated in Barth's solution and studied 1–4 days after injection.

**Electrophysiology**. Inside-out patch-clamp recordings were performed using an EPC7 amplifier (List Electronik) at a constant holding potential of −60 mV. The pipette solution contained 140 mM KCl, 1.2 mM $MgCl_2$, 2.6 mM $CaCl_2$, and 10 mM HEPES (pH 7.4 with KOH). For experiments with neomycin, the Mg-free intracellular solution contained 107 mM KCl, 10 mM HEPES, and 10 mM EDTA (pH 7.2 with KOH). To account for possible rundown, the control current ($I_C$) was taken as the mean of the current in control solution before and after neomycin application. Concentration–response curves were fitted with a modified Hill Eq. (1):

$$\frac{I}{I_c} = a + \frac{(1-a)}{1 + \left(\frac{[X]}{IC_{50}}\right)^h} \qquad (1)$$

where $[X]$ is the concentration of the test substance, $IC_{50}$ is the concentration at which inhibition is half maximal, $h$ is the slope factor (Hill coefficient) and $a$ represents the fraction of unblocked current at high neomycin concentrations when binding saturation occurs . Single-channel currents were recorded at −60 mV, filtered at 5 kHz, sampled at 20–50 kHz, and analysed using a combination of Clampfit (Axon Instruments) and GraphPad Prism 8. Data are given as mean ± SEM.

**Reporting summary**. Further information on research design is available in the Nature Research Reporting Summary linked to this article.

## Data availability
All data are available from the corresponding author on request.

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

## Acknowledgements

We thank Mark Sansom for fruitful discussions and Irfan Alibay, Josh Sauer, and Owen Vickery for advice and technical support. T.P. holds a Wellcome Trust OXION studentship and a Clarendon scholarship. Research in P.J.S.'s lab is funded by Wellcome (208361/Z/17/Z), the MRC (MR/S009213/1), and BBSRC (BB/P01948X/1, BB/R002517/1, and BB/S003339/1). Research in F.M.A.'s lab is funded by the MRC (MR/T002107/1), BBSRC (BB/R002517/1, BB/R017220/1), and Wellcome Trust (102161/Z/13/Z). This project made use of time on ARCHER and JADE granted via the UK High-End Computing Consortium for Biomolecular Simulation, HECBioSim (http://hecbiosim.ac.uk), supported by EPSRC (Grant number EP/R029407/1). P.J.S. acknowledges Athena at HPC Midlands+, which was funded by the EPSRC on grant EP/P020232/1, and the University of Warwick Scientific Computing Research Technology Platform for computational access.

## Author contributions

T.P. prepared the mutants for the electrophysiological studies and performed the coarse-grained molecular dynamics simulations and free energy calculations. P.P. provided recording from the electrophysiology experiments. T.P., R.A.C., F.M.A., and P.J.S. jointly designed the experiments, analysed the data, and wrote the manuscript.

## Competing interests

The authors declare no competing interests.
