## [Peer Review File · Communications Chemistry]

Reviewers' comments:

Reviewer #1 (Remarks to the Author):

This manuscript reports on free energy calculations and functional studies to investigate interactions between PIP2 and Kir channels. It is well-written and straightforward to read, and the calculations appear to be well done. To my knowledge, it would be the first published application of FEP toward quantifying lipid specificity, which is a very useful contribution. My largest concern is with a sparse amount of larger context for the work, which also makes it more challenging to evaluate significance.

1. Background: The larger application of the work considers the mechanisms underlying two disease-causing mutations. The existing "knowledge gap" in the mechanism is not clarified in the introduction - what previous work has been done to understand the mechanism, and how do the results of the authors build on it? In general, the introduction is sparse on references to previous work on Kir channels.

2. FEP: The authors claim in the discussion that "we demonstrate that amino acid mutations can be investigated using FEP", but mutation of amino acids (of soluble proteins within atomistic simulations) is actually one of the oldest applications of FEP. See, for example, exercise 3 in the tutorial at <https://www.ks.uiuc.edu/Training/Tutorials/namd/FEP/tutorial-FEP.pdf>. In my opinion it is an underused approach and the authors applied it well, but the larger context needs to be clear.

3. CG-FEP:

a) The use of a coarse-grained forcefield here is understandable but the limitations need to be made explicit. The coarse-grained forcefield introduces a loss of precision that I think is acceptable when the measured free energy differences are large (it would be very surprising if a free energy difference of 20-30 kJ/mol vanished using an atomistic simulation; if true, the coarse-grained community would need to know immediately) although I would still argue this point explicitly.

I am more concerned that the CG approach will underestimate moderate free energy differences and give "false negatives". For instance, the authors observe a "very small free energy change" for transforming PIP2 to PI4P, but the coarse-graining process reduces the differences in shape, charge distribution, etc between chemical analogues. The potential for underestimating relative free energy differences should be clarified.

b) Fig 2b will probably surprise readers who aren't familiar with CG forcefields. Please include a figure showing the Martini mapping for all four lipids.

c) The authors should probably consider whether any surprising and unexplained results could be related to use of the CG-FF (for instance, the surprising result for hKir3.2).

4. Effect of bulk membrane composition: The bulk membrane calculations were all done in POPC. A natural hypothesis is that these values could change considerably in a membrane containing anionic phospholipids. Testing that hypothesis would provide important perspective on the results. I request that the authors check whether the transformation shown in Figure 2b or c is affected by introducing 10-20% anionic phospholipids into the bulk membrane.

Minor concerns:

1. Fig 1 b-d: The y axis limits unnecessarily compress the data. Please reduce the maximum (and for c and d I'd also increase the minimum.)

2. Fig 2b : There is a stray "5" to the left side of the figure

3. Fig 5d : The minimally-annotated sequence alignment is hard to read. I suggest introducing secondary structure annotation and cropping out the long stretches that don't interact with the lipid in any of the channels. Coloring by charge or annotating conservation could also help with

navigation.

4. The writing in the last paragraph of the results is difficult to parse. This manuscript reports on free energy calculations and functional studies to investigate interactions between PIP2 and Kir channels. It is well-written and straightforward to read, and the calculations look to be done well. My largest concern is with a sparse amount of larger context for the work, which also makes it more challenging to evaluate significance.

1. Background: The larger application of the work concerns the mechanisms underlying two disease-causing mutations. The existing "knowledge gap" in the mechanism is not clarified in the introduction - what previous work has been done to understand the mechanism, and how do the results of the authors build on it?

2. FEP: The authors claim in the discussion that "we demonstrate that amino acid mutations can be investigated using FEP", but mutation of amino acids (of soluble proteins within atomistic simulations) is actually one of the oldest applications of FEP. See, for example, exercise 3 in the tutorial at <https://www.ks.uiuc.edu/Training/Tutorials/namd/FEP/tutorial-FEP.pdf>. In this reviewer's opinion it is an underused approach and the authors applied it well, but the larger context needs to be clear.

3. CG-FEP:

a) The use of a coarse-grained forcefield here is understandable but the limitations need to be made explicit. The coarse-grained forcefield introduces a loss of precision that I think is acceptable when the measured free energy differences are large (it would be very surprising if a free energy difference of 20-30 kJ/mol vanished using an atomistic simulation; if true, the coarse-grained community would need to know immediately) although this still needs to be addressed.

I am more concerned that the CG approach will underestimate moderate free energy differences and give "false negatives". For instance, the authors observe a "very small free energy change" for transforming PIP2 to PI4P, but the coarse-graining process automatically reduces the differences in shape, charge distribution, etc between chemical analogues. The potential for underestimating relative free energy differences should be clarified.

b) Fig 2b will probably surprise readers who aren't familiar with CG forcefields. Please include a figure showing the Martini mapping for all four lipids.

c) Do the authors have any reason to believe the surprising but consistent result for hKir3.2 could be related to use of the CG-FF?

4. Effect of bulk membrane composition: The bulk membrane calculations were all done in POPC. I hypothesize that these values could change considerably in a membrane containing anionic phospholipids, and if so, this would be both valuable information and provide important perspective on the results. I request that the authors test whether the transformation shown in Figure 2b or c is affected by introducing 10-20% anionic phospholipids into the bulk membrane.

Minor concerns:

1. Fig 1 b-d: The y axis limits unnecessarily compress the data. Please reduce the maximum (and for c and d I'd also increase the minimum.)

2. Fig 2b : There is a stray "5" to the left side of the figure

3. Fig 5d : The minimally-annotated sequence alignment is hard to read. I suggest introducing secondary structure annotation and cropping out the long stretches that don't interact with the lipid in any of the channels. Coloring by charge or annotating conservation could also help with navigation.

4. The writing in the last paragraph of the results is difficult to parse.

Reviewer #2 (Remarks to the Author):

Paper: Evaluating Inositol phospholipid interactions with Inward Rectifier Potassium Channels and their role in Disease

Protein-lipid interactions are very heavy nowadays in research focus. The subject will be of interest to others. Many research teams are exploring these type interactions for example the excellent paper from de Groot in the same journal: <https://www.nature.com/articles/s41467-020-15741-8>

K⁺ channels are well-known for their involvement in disease and Pr. Ashcroft is a leader on this.

In this paper the team claim to develop a new method and to have established a platform. This will be very interesting news to the people who really developed perturbation free energy methods. The words used are very strong & one example is: 'power of our approach'. The investigators have used free energy perturbation method that researchers in the simulation field have been using for numerous years- I advice to please keep the words at the appropriate level.

The investigators should choose if this paper is about methods development or a paper that gives novel results of protein-lipid binding because as it is written it is neither of these things.

Computationally, this is not a heavy study. 1 microsecond length simulations with the coarse-grained force-field they have used can be achieved on a modest workstation in only a few days (maybe less). They have generated a few models, completed some real structures and then done some short CG MD simulations. It is disappointing. If the focus will be on methods the investigators should show different lipids which are similar to PIP2 binding in the same sites and report if the energy calculated will show differences between the lipids. All atom conventional force-field simulations should be included for comparison to really prove about the coarse-grain numbers.

The title is somehow misleading. The disease part should be removed as it is not the focus and has not been shown in many details in the paper.

The electrophysiology is done with standard methods, this part seems good.

The references are good.

In my opinion this paper would benefit from refocusing, rewording and providing more convincing data about the lipid binding before it can be seriously considered.

Reviewer #3 (Remarks to the Author):

The manuscript by Stansfeld et al. describes simulations and experiments on the Kir6.2 and related potassium channel, aiming at understanding the energetics of binding of PIP2 lipids to the protein and to a few pathological mutants. The subject is certainly interesting for a biochemistry audience. The manuscript reads well and the figures are clear. Different from what is claimed by the authors, the computational methods are well-established, and there is no significant novelty from the methodological standpoint. On the other hand, the simulation results are novel, to the best of my knowledge.

I list below some minor points to be addressed.

1. I found it difficult to understand the actual structure of the complex Kir6.2/SUR1 in the membrane, as no figure is provided for it. If SUR1 is around Kir6.2, how can PIP2 access Kir6.2? A figure would clarify.

2, Page 21, the Discussion section starts with: "Here we present a novel method for comparing the binding free energy between two lipid species to a given site on a membrane protein, based on previously developed free energy perturbation (FEP) calculations [3]."

I do not see any new method developed here, but just application of well-established methods. Second, the sentence is ambiguous, as it seems to imply that FEP methods were developed in ref. 3, which is certainly not the case. Instead, a proper citation should be provided for FEP methods.

3. The Discussion is very limited, both with respect to methods and the biological relevance of the results. At least for the biological relevance, a more careful discussion of experimental results obtained by others would help understanding the relevance of the particular protein studied here.

4. The Martini protein force field is used in all simulations, but it is not cited.

5. A citation for the stochastic V-rescale thermostat is also missing.

6. Overall, the simulation length is unimpressive: CG modeling actually allows large scale simulations, while sub-microsecond time scales (reported here for FEP calculations) are easily accessible with more accurate all-atom MD simulations. FEP-CG calculations with sampling better than reported here were already published over a decade ago. I find it surprising that statistical uncertainties reported are so low, considering the relatively short sampling.

7. The simulation setup is rather poorly explained in the methods section, and does not allow reproducing the calculations. How many POPC and PIP2 lipids in the unbiased simulations? Water molecules, ions? How were all-atom structures converged to CG structures? FEP: PIP3 lipids are mentioned in Methods at p. 22, but not in the article text (typo?). The values of the lambda parameter and soft-core parameters are not specified. A phosphate group is PO₄, not PO₃.

My comments are biased by the unavailability of the Supporting Info file (the one provided is unreadable for me).

Reviewers' comments:

Reviewer #1 (Remarks to the Author):

This manuscript reports on free energy calculations and functional studies to investigate interactions between PIP₂ and Kir channels. It is well-written and straightforward to read, and the calculations appear to be well done. To my knowledge, it would be the first published application of FEP toward quantifying lipid specificity, which is a very useful contribution. My largest concern is with a sparse amount of larger context for the work, which also makes it more challenging to evaluate significance.

We thank the reviewer for the acknowledgement of the contribution to the field. We have contextualised the work, as detailed below, and expanded the introduction to capture this.

1. Background: The larger application of the work considers the mechanisms underlying two disease-causing mutations. The existing "knowledge gap" in the mechanism is not clarified in the introduction - what previous work has been done to understand the mechanism, and how do the results of the authors build on it? In general, the introduction is sparse on references to previous work on Kir channels.

We have now expanded the introduction to highlight how the channel's mutations are associated with neonatal diabetes and how previous studies have addressed the channelopathy aspects based on a) functional studies of the E179A mutation (Flanagan et al., 2007 and Haider et al., 2007) and b) Lack of channel activation by the K67N mutation (Reimann et al., 2003). Prior to this study it had not been defined whether the mutation affects binding or channel activation by PIP₂ and hence, this adds the significance to the work detailed here.

2. FEP: The authors claim in the discussion that "we demonstrate that amino acid mutations can be investigated using FEP", but mutation of amino acids (of soluble proteins within atomistic simulations) is actually one of the oldest applications of FEP. See, for example, exercise 3 in the tutorial at <https://www.ks.uiuc.edu/Training/Tutorials/namd/FEP/tutorial-FEP.pdf>. In my opinion it is an underused approach and the authors applied it well, but the larger context needs to be clear.

We have modified our text accordingly as the wording conveys a different meaning to what we intended. As the reviewer points out we have not invented FEP for single amino acid mutations. We have added a supporting paragraph to the introduction (paragraph 4) to acknowledge previous work done on the mutation of an atomistic FEP in the introduction (ref 19-21). We have added the word "coarse-grained" to the sentence to clarify the aim of doing FEP at a much cheaper computational cost and acknowledge previous methodologies in the second paragraph in the discussion.

3. CG-FEP:

a) The use of a coarse-grained forcefield here is understandable but the limitations need to be made explicit. The coarse-grained forcefield introduces a loss of precision that I think is acceptable when the measured free energy differences are large (it would be very surprising if a free energy difference of 20-30 kJ/mol vanished using an atomistic simulation; if true, the coarse-grained community would need to know immediately) although I would still argue this point explicitly.

We have added a sentence to acknowledge that the value at approximately 20-30 kJ/mol is acceptable for a CG-based energy calculation and the limitations of using CG versus an atomistic-level approach. This has been illustrated through previous studies on PIP₂ binding to Kir2.2 (ref 35-37), and other PIP binding proteins. All additional citations are added to the first paragraph in the discussion.

I am more concerned that the CG approach will underestimate moderate free energy differences and give "false negatives". For instance, the authors observe a "very small free energy change" for transforming PIP₂ to PI4P, but the coarse-graining process reduces the differences in shape, charge distribution, etc between chemical analogues. The potential for underestimating relative free energy differences should be clarified.

We have commented on the impact of underestimating the free energy in our discussion. However, the sums of all step perturbations show a very promising closed thermodynamics cycle, and we have shown that our results agree very well within three independent simulation repeats.

b) Fig 2b will probably surprise readers who aren't familiar with CG forcefields. Please include a figure showing the Martini mapping for all four lipids.

We have added Figure 2b - the mapping of the CG to atomistic and have shifted other figures down (2b → 2c) etc.

c) The authors should probably consider whether any surprising and unexplained results could be related to use

of the CG-FF (for instance, the surprising result for hKir3.2).

We have acknowledged the caveats of the MARTINI forcefield in the text to address the surprising results of the hKir3.2 at the very end of the last paragraph in the results section.

4. Effect of bulk membrane composition: The bulk membrane calculations were all done in POPC. A natural hypothesis is that these values could change considerably in a membrane containing anionic phospholipids. Testing that hypothesis would provide important perspective on the results. I request that the authors check whether the transformation shown in Figure 2b or c is affected by introducing 10-20% anionic phospholipids into the bulk membrane.

We have addressed this question in the supplementary figure 7. We show that conducting the simulation in 10% PS does not affect the $\Delta\Delta G$ value (n=3).

Minor concerns:

1. Fig 1 b-d: The y axis limits unnecessarily compress the data. Please reduce the maximum (and for c and d I'd also increase the minimum.)

Fig 1b: We have decreased the maximum from 2.5 to 2 Å

Fig 2c-2d: We have decreased the maximum from 14 to 10 Å, and increased the minimum to 4 Å

2. Fig 2b : There is a stray "5" to the left side of the figure

We have removed the stray 5 from the figure.

3. Fig 5d : The minimally-annotated sequence alignment is hard to read. I suggest introducing secondary structure annotation and cropping out the long stretches that don't interact with the lipid in any of the channels. Coloring by charge or annotating conservation could also help with navigation.

We have annotated the secondary structure on the sequence alignment.

We have cropped out regions which the binding are not highly conserved between all four channels.

We have colour coded the residues, red - acidic, blue - basic, green - polar and black - hydrophobic to help the reader.

4. The writing in the last paragraph of the results is difficult to parse.

We have re-written the last paragraph of the results section.

This manuscript reports on free energy calculations and functional studies to investigate interactions between PIP2 and Kir channels. It is well-written and straightforward to read, and the calculations look to be done well. My largest concern is with a sparse amount of larger context for the work, which also makes it more challenging to evaluate significance.

We thank the reviewer for their comments, which have enhanced the quality of our manuscript.

Reviewer #2 (Remarks to the Author):

Paper: Evaluating Inositol phospholipid interactions with Inward Rectifier Potassium Channels and their role in Disease

Protein-lipid interactions are very heavy nowadays in research focus. The subject will be of interest to others. Many research teams are exploring these type interactions for example the excellent paper from de Groot in the same journal: <https://www.nature.com/articles/s41467-020-15741-8>

We thank the reviewer for the acknowledgement of the contribution to the field and their suggestion of this recent paper from Nature Communications. We have now cited the paper in our introduction (ref. 2).

K⁺ channels are well-known for their involvement in disease and Pr. Ashcroft is a leader on this.

In this paper the team claim to develop a new method and to have established a platform. This will be very interesting news to the people who really developed perturbation free energy methods. The words used are very strong & one example is: 'power of our approach'. The investigators have used free energy perturbation method that researchers in the simulation field have been using for numerous years- I advice to please keep the words at the appropriate level.

As noted above, it was not our intention to convey that we had developed FEP calculations, rather we have applied FEP calculations in the context of CG simulations. We have toned down the language throughout the manuscript as follows:

Introduction : Paragraph 4 : We add an additional paragraph to acknowledge previous work done using FEP and change the wording in the sixth paragraph from the "power of our approach" to "our methodological application of the CG-FEP describes" to tone down the language.

Discussion: We have also addressed the significance of the previous FEP methods in the second paragraph in the discussion section and how coarse-grained FEP could be contributing to the field as an extension to a previously developed method.

The investigators should choose if this paper is about methods development or a paper that gives novel results of protein-lipid binding because as it is written it is neither of these things.

Ultimately, this is a problem-based application-focussed manuscript, rather the development of a methodology and therefore we have stripped back the method development side of the paper. The paper has outlined the potential of extending the widely-used FEP approach by applying coarse-grained simulation in order to address biochemical questions.

Computationally, this is not a heavy study. 1 microsecond length simulations with the coarse-grained force-field they have used can be achieved on a modest workstation in only a few days (maybe less). They have generated a few models, completed some real structures and then done some short CG MD simulations. It is disappointing.

We acknowledge the short simulation timescale. However, as shown in Figure 1b, all our simulations are well converged within 1 μ s simulation. As the simulation has converged (PIP₂ has bound stably in the binding site), we have decided to approach "more replicates" than an extension of a simulation as we are not interested in the dissociation constant of the binding. This yields a total of 5 μ s simulation per ion channel. The majority of the simulation cost lies in the FEP calculation as follows:

Simulations in bulk POPC bilayer: 300ns x 21 simulations windows x 5 conditions (PIP₂ → PI4P, PI4P → PI, PI → POPC. PIP₂ → POPC, PIP₂ → diC8:PIP₂) x 3 repeats = 94.5 μ s

Transformation involving Kir channels without the flat bottom restraint: 300ns x 21 simulations windows x 4 conditions (PIP₂ → PI4P, PI4P → PI, PI → POPC. PIP₂ → POPC) x 3 repeats x 4 ion channels (hKir1.1, hKir2.2, hKir3.2 and hKir6.2) = 302.4 μ s

Transformation involving Kir channels with the flat bottom restraint: 300ns x 21 simulations windows x 3 conditions (PIP₂ → PI4P, PI → POPC. PIP₂ → POPC) x 3 repeats x 1 ion channels (hKir6.2) = 56.7 μ s

Transformation of an amino acids without PIP₂: 300ns x 21 simulations windows x 4 conditions (E179K, E179A, K67N, C166S) x 3 repeats = 75.6 μ s

Transformation of an amino acids with PIP₂: 300ns x 21 simulations windows x 4 conditions (E179K, E179A, K67N, C166S) x 3 repeats = 75.6 μ s

Transformation of diC8:PIP₂ with Kir6.2: 300ns x 21 simulations x 3 repeats = 18.9 μ s

Transformation involving Kir channels in the presence of SUR1: 300ns x 21 simulations windows from PIP₂ → POPC x 3 repeats = 18.9 μ s

PMF calculation (500 ns x 40 simulations window) = 20 μ s

This add up to a total simulation cost of 20+94.5+302.4+56.7+75.6+75.6+18.9+18.9+20 = 682.6 μ s, which we have now added to the materials and method section to relay this to the reader. With our relatively powerful workstation (i9-10980X with RTX2080 GPU), we could simulate our system at approximately 3 μ s per day. Thus, the total time required for all simulations to be completed, excluding energy minimisation and equilibration, would be approximately 227 days on a single node.

If the focus will be on methods the investigators should show different lipids which are similar to PIP₂ binding in the same sites and report if the energy calculated will show differences between the lipids. All atom conventional force-field simulations should be included for comparison to really prove about the coarse-grain numbers.

As previously mentioned, we aim to apply coarse-grained method to address biochemical phenomena and to apply the CG-FEP method to characterise PIP₂ binding. Our perturbation also involves charge species which is currently difficult to obtain accurate free energies and convergence for an atomistic system, hence why we use CG simulations to provide a measurement for the affinity of PIP₂ binding.

The title is somehow misleading. The disease part should be removed as it is not the focus and has not been shown in many details in the paper.

As we have rejigged the manuscript to focus on the application, the disease aspect becomes more central to the study. It is discussed in two figures and in both experimental and computational studies. We feel that the title is a fair reflection of the paper.

The electrophysiology is done with standard methods, this part seems good.

We thank the reviewer for acknowledging the electrophysiology section of the paper.

The references are good.

We thank the reviewer for acknowledging the reference section of the paper.

In my opinion this paper would benefit from refocusing, rewording and providing more convincing data about the lipid binding before it can be seriously considered.

We have toned down the methodology section of the paper to refocus and reword as suggested.

We thank the reviewer for their comments, which have enhanced the quality our manuscript.

Reviewer #3 (Remarks to the Author):

The manuscript by Stansfeld et al. describes simulations and experiments on the Kir6.2 and related potassium channel, aiming at understanding the energetics of binding of PIP2 lipids to the protein and to a few pathological mutants. The subject is certainly interesting for a biochemistry audience. The manuscript reads well and the figures are clear.

We thank the reviewer for the acknowledgement of the contribution to the field and the clarity of the figures.

Different from what is claimed by the authors, the computational methods are well-established, and there is no significant novelty from the methodological standpoint. On the other hand, the simulation results are novel, to the best of my knowledge.

We have refocussed the manuscript as a more application-based approach and an extension to the previous method as follows:

Introduction : Paragraph 4 : We added an additional paragraph to acknowledge previous work done using FEP and changed the wording in the sixth paragraph from the "power of our approach" to "our methodological application of the CG-FEP describes" to tone down the language.

Discussion : We have also addressed the significance of the previous FEP methods in the second paragraph in the discussion section and addressed how coarse-grained FEP could be contributing to the field as an extension to a previously developed method.

The paper has provided an extension to the previous approach (Atomistic FEP) using coarse-grained simulation in order to address biological questions. We thank the review for acknowledging the novelty of the simulation results.

I list below some minor points to be addressed.

1. I found it difficult to understand the actual structure of the complex Kir6.2/SUR1 in the membrane, as no figure is provided for it. If SUR1 is around Kir6.2, how can PIP2 access Kir6.2? A figure would clarify.

We have placed the structure of Kir6.2+SUR1 with PIP₂ in the binding site to clarify this point in the figure 4c.

2, Page 21, the Discussion section starts with: "Here we present a novel method for comparing the binding free energy between two lipid species to a given site on a membrane protein, based on previously developed free energy perturbation (FEP) calculations [3]."

I do not see any new method developed here, but just application of well-established methods.

We have added the word “coarse-grained” to the sentence to clarify the aim of doing FEP at a much cheaper computational cost as an application to the previous method. We have also added a supporting sentence to acknowledge previous work done on the mutation of an atomistic FEP in the discussion.

Second, the sentence is ambiguous, as it seems to imply that FEP methods were developed in ref. 3, which is certainly not the case. Instead, a proper citation should be provided for FEP methods.

We have also added a supporting paragraph to the introduction (paragraph 4) to acknowledge previous work done on the mutation of an atomistic FEP in the introduction (ref 19-21). We have added the word “coarse-grained” to the sentence to clarify the aim of doing FEP at a much cheaper computational cost and acknowledge previous methodologies in the second paragraph in the discussion.

3. The Discussion is very limited, both with respect to methods and the biological relevance of the results. At least for the biological relevance, a more careful discussion of experimental results obtained by others would help understanding the relevance of the particular protein studied here.

Unfortunately, *in vitro* lipid binding studies to the Kir channel has been done to a very limited extent. We have added Fan and Makielski (1997) which shows activation of cardiac K_{ATP} channel by different phosphoinositides to support our binding studies.

4. The Martini protein force field is used in all simulations, but it is not cited.

We have now cited Marrink et al (2007) in the methods section.

5. A citation for the stochastic V-rescale thermostat is also missing.

We have now cited Bussi et al (2007) for v-rescale thermostat.

6. Overall, the simulation length is unimpressive: CG modeling actually allows large scale simulations, while sub-microsecond time scales (reported here for FEP calculations) are easily accessible with more accurate all-atom MD simulations. FEP-CG calculations with sampling better than reported here were already published over a decade ago. I find it surprising that statistical uncertainties reported are so low, considering the relatively short sampling.

We acknowledge the short simulation timescale. However, as shown in Figure 1b, all our simulations are well converged within 1 μ s simulation. As the simulation has converged (PIP₂ has bound stably in the binding site), we have decided to approach “more replicates” than an extension of a simulation as we are not interested in the dissociation constant of the binding. This yields a total of 5 μ s simulation per ion channel. The majority of the simulation cost lies in the FEP calculation as follows:

We acknowledge the short simulation timescale. However, as shown in Figure 1b, all our simulations are well converged within 1 μ s simulation. As the simulation has converged (PIP₂ has bound stably in the binding site), we have decided to approach “more replicates” than an extension of a simulation as we are not interested in the dissociation constant of the binding. This yields a total of 5 μ s simulation per ion channel. The majority of the simulation cost lies in the FEP calculation as follows:

Simulations in bulk POPC bilayer: 300ns x 21 simulations windows x 5 conditions (PIP₂ → PI4P, PI4P → PI, PI → POPC. PIP₂ → POPC, PIP₂ → diC8:PIP₂) x 3 repeats = 94.5 μ s

Transformation involving Kir channels without the flat bottom restraint: 300ns x 21 simulations windows x 4 conditions (PIP₂ → PI4P, PI4P → PI, PI → POPC. PIP₂ → POPC) x 3 repeats x 4 ion channels (hKir1.1, hKir2.2, hKir3.2 and hKir6.2) = 302.4 μ s

Transformation involving Kir channels with the flat bottom restraint: 300ns x 21 simulations windows x 3 conditions (PIP₂ → PI4P, PI → POPC. PIP₂ → POPC) x 3 repeats x 1 ion channels (hKir6.2) = 56.7 μ s

Transformation of an amino acids without PIP₂: 300ns x 21 simulations windows x 4 conditions (E179K, E179A, K67N, C166S) x 3 repeats = 75.6 μ s

Transformation of an amino acids with PIP₂: 300ns x 21 simulations windows x 4 conditions (E179K, E179A, K67N, C166S) x 3 repeats = 75.6 μ s

Transformation of diC8:PIP₂ with Kir6.2: 300ns x 21 simulations x 3 repeats = 18.9 μ s

Transformation involving Kir channels in the presence of SUR1: 300ns x 21 simulations windows from PIP₂ → POPC x 3 repeats = 18.9 μ s

PMF calculation (500 ns x 40 simulations window) = 20 μ s

This add up to a total simulation cost of $20+94.5+302.4+56.7+75.6+75.6+18.9+18.9+20 = 682.6 \mu$ s, which we have now added to the materials and method section to relay this to the reader. With our relatively powerful workstation (i9-10980X with RTX2080 GPU) we could simulate our system at approximately 3 μ s per day. Thus, the total time required for all simulations to be completed excluding energy minimisation and equilibration would be approximately 227 days on a single node. This does not capture the need to establish the equilibrated systems and account for analysis and interpretation of the results. Ultimately the length of simulations are suitable for full convergence and relatively low error.

7. The simulation setup is rather poorly explained in the methods section, and does not allow reproducing the calculations. How many POPC and PIP2 lipids in the unbiased simulations? Water molecules, ions?

We have now added a table of simulations set-up to support our method section as supplementary table 8.

How were all-atom structures converged to CG structures?

We have cited a self-assembly MemProtMD pipeline and explain the full detail of the pipeline, allowing calculation to be reproducible by others.

FEP: PIP3 lipids are mentioned in Methods at p. 22, but not in the article text (typo?).

Thank you. We have now corrected the typos throughout the manuscript.

The values of the lambda parameter and soft-core parameters are not specified.

We have now provided that information in the materials and method section.

A phosphate group is PO₄, not PO₃.

Thank you. This was taken from the naming convention in the Martini forcefield parameters, rather than us being unaware that phosphate is PO₄. We have now corrected the information throughout the manuscript.

My comments are biased by the unavailability of the Supporting Info file (the one provided is unreadable for me).

We have now forwarded the Supporting Info file to the reviewer.

We thank the reviewer for their comments, which have enhanced the quality our manuscript.

Reviewers' comments:

Reviewer #1 (Remarks to the Author):

I am satisfied with the changes made.

Reviewer #2 (Remarks to the Author):

The paper is now improved a lot. I still think it suffers from confusion about methods versus applications. 'Our methodology' is used - it is just a methodology, not your methodology, using CG with FEP does not make it a new methodology. It is your application of an existing methodology. This really should be addressed

The lipid specificity is potentially very interesting and significant. But my point about the difference between PIP lipids and other similar ones has not been addressed. Instead the authors neglect to answer the questions. This is an important point for the paper as the selective binding of lipids is being studied. There should be clear evidence that the CG force field can distinguish between lipids that are similar.

Reviewer #3 (Remarks to the Author):

The authors re-wrote a substantial part of the manuscript, that now is more focused on the application of the FEP technique to an interesting case study. In my view, they addressed most of the referees concerns in a satisfactory way.

Reviewers' comments:

Reviewer #1 (Remarks to the Author):

I am satisfied with the changes made.

We are pleased that Reviewer #1 is now happy with the changes we have made.

Reviewer #2 (Remarks to the Author):

The paper is now improved a lot. I still think it suffers from confusion about methods versus applications. 'Our methodology' is used - it is just a methodology, not your methodology, using CG with FEP does not make it a new methodology. It is your application of an existing methodology. This really should be addressed

We have change "our method" to "the method" in the abstracts, introductions, results and discussion, and clarified this throughout the manuscript.

The lipid specificity is potentially very interesting and significant. But my point about the difference between PIP lipids and other similar ones has not been addressed. Instead the authors neglect to answer the questions. This is an important point for the paper as the selectively binding of lipids is being studied. There should be clear evidence that the CG force field can distinguish between lipids that are similar.

We highlight through the transformation of one PIP₂ to another that we can distinguish between lipids species. We have cited previous work which show the use of the CG forcefield in distinguishing two PIP lipid species, eg:

One of our concerns is the ability of the CG forcefield to distinguish between similar inositol lipids in the free energy calculations. This application of CG-FEP has demonstrated that the method can show differences between PIP₂, PI4P, PI and PC at the accuracy of at least 5 kJ/mol (~1.5 k_BT)³⁷. This complements the previous free energy calculations which has shown that CG forcefield is able to distinguish between PIP₂ and PIP₃ lipids binding to a PH domain.

Reviewer #3 (Remarks to the Author):

The authors re-wrote a substantial part of the manuscript, that now is more focused on the application of the FEP technique to an interesting case study. In my view, they addressed most of the referees concerns in a satisfactory way.

We thank the reviewer for their positive comments.

REVIEWERS' COMMENTS:

Reviewer #2 (Remarks to the Author):

I am now satisfied with the manuscript. The appropriate rigour and scholarly details have been applied, in response to the reviewers comments.